# Disease Severity and Comorbidities among Healthcare Worker COVID-19 Admissions in South Africa: A Retrospective Analysis

**DOI:** 10.3390/ijerph19095519

**Published:** 2022-05-02

**Authors:** Edith Ratshikhopha, Munyadziwa Muvhali, Nisha Naicker, Nonhlanhla Tlotleng, Waasila Jassat, Tanusha Singh

**Affiliations:** 1National Health Laboratory Service, National Institute for Occupational Health, Johannesburg 2000, South Africa; edithr@nioh.ac.za (E.R.); munyadziwam@nioh.ac.za (M.M.); nonhlanhlat@nioh.ac.za (N.T.); 2Department of Clinical Microbiology and Infectious Diseases, School of Pathology, University of the Witwatersrand, Johannesburg 2000, South Africa; 3Department of Environmental Health, University of Johannesburg, Doornfontein 2028, South Africa; nnaicker@uj.ac.za; 4National Health Laboratory Service, National Institute for Communicable Diseases, Sandringham 2192, South Africa; waasilaj@nicd.ac.za

**Keywords:** coronavirus, risk factors, comorbidities, vulnerable, disease severity, health outcomes

## Abstract

Healthcare workers (HCWs) are among the most vulnerable in regard to contracting severe acute respiratory syndrome coronavirus 2 (SARS-CoV-2) infection. Comorbidities are reported to increase the risk for more severe COVID-19 outcomes, often requiring hospitalization. However, the evidence on disease severity and comorbidities among South African HCWs is lacking. This retrospective study analyzed the prevalence of comorbidities among HCW hospitalized with COVID-19 and its association with the severity of outcomes. Data from public and private hospitals in nine provinces of South Africa were extracted from the national hospital surveillance database for COVID-19 admissions. A total of 10,149 COVID-19 HCWs admissions were reported from 5 March 2020 to 31 December 2021. The risk of disease severity among HCWs increased with age, with those older (≥60 years) having seven times the odds of disease severity (aOR 7.0; 95% CI 4.2–11.8) compared to HCWs in the younger age (20–29 years) group. The most commonly reported comorbidity was hypertension (36.3%), followed by diabetes (23.3%) and obesity (16.7%). Hypertension (aOR 1.3; 95% CI 1.0–1.6), diabetes (aOR 1.6; 95% CI 1.3–2.0), and HIV (aOR 1.6; 95% CI 1.2–2.1) were significantly associated with disease severity. In conclusion, age, gender, and existing comorbidities were strong predictors of the prognosis of severe COVID-19 among HCWs in South Africa. The information is important in the development of occupational health policies and vulnerability risk assessments for HCWs in light of future COVID-19 waves or similar outbreaks.

## 1. Introduction

Healthcare workers (HCWs) are a high-risk group for infections, including tuberculosis (TB) [1,2], and are among the most vulnerable to severe acute respiratory syndrome coronavirus 2 (SARS-CoV-2) infection [3]. Healthcare workers are in the frontline of the coronavirus disease of 2019 (COVID-19) outbreak response and are the most at risk of acquiring the infection, as they are exposed to infectious material [4,5,6]. A recent rapid review reported a lack of and/or inadequate personal protection equipment (PPE), exposure to infected patients, work overload, poor infection control, and pre-existing medical conditions as important risk factors for COVID-19 infection among HCWs [7]. As with many the HCWs in other countries, South African HCWs have not been spared from the coronavirus disease of 2019 (COVID-19); and they are reported to be at higher risk of getting infected [8,9,10]. In Europe, the overall positivity amongst HCWs was 8.8% in Milan, Italy, with a positivity rate of 24% in symptomatic HCWs alone [11]. However, a much higher prevalence of 42.4% of SARS-CoV-2 infection among symptomatic HCWs was reported in a large university tertiary hospital in São Paulo, Brazil [12]. In a cross-sectional study in Egypt, a 14.3% PCR positivity was reported among asymptomatic HCWs [13]. Similar rates (14%) were reported among primary health HCWs of HIV service providers in five districts of South Africa by the end of September 2020 [14]. In Cape Town, South Africa, 10.4% seroprevalence among HCWs from pediatric facilities enrolled between May and July 2020 were reported [15]. In another South African study, 34.6% of HCWs in a cohort from a large tertiary hospital had PCR-confirmed SARS-CoV-2. Among PCR-confirmed cases, 59.3% were symptomatic, 30.4% were pre-symptomatic, and 10.4% remained asymptomatic [6].

Studies, including a systematic review and meta-analysis, showed that severe COVID-19 and mortality were associated with gender, age, smoking status, and comorbidities. The reported comorbidities included obesity, hypertension, cardiovascular diseases, diabetes, chronic obstructive pulmonary disease (COPD), asthma, and cancer [16,17,18,19,20,21,22,23]. Additionally, the age of the patients has also been a contributing factor [17,19,20,21,22,23,24,25]. Most of these studies reporting an association of COVID-19 and comorbidities were from Asia, Europe, and America [25,26,27,28,29,30,31,32,33] and three representing Africa. A South African study investigated the role of comorbidities and health outcome among all COVID-19 hospital admissions [34]. In addition, the Critical Care Outcomes Study (ACCCOS), a multicenter cohort study in 64 hospitals from ten African countries, including South Africa (SA), reported an association between increased mortality and HIV/AIDS, diabetes, chronic liver disease, kidney disease, and severity of organ dysfunction at admission of adults with suspected or confirmed COVID-19, as well as insufficient critical care resources [35]. However, no significant association was found between COVID-19 infection with chronic lung disease and active TB. The third African study conducted between March and June 2020 in South African district hospitals in the Western Cape province also did not show an association between increased mortality and TB, asthma, chronic obstructive pulmonary disease, and post-TB structural lung disease among COVID-19 positive patients. Mortality was significantly associated with increasing age, male sex, obesity, type 2 diabetes, HIV, and chronic kidney disease [36]. A total of 126,226 COVID-19 infections (2.1% of all cases) among health workers in the region were reported, with South Africa accounting for about 45.0% (56,180) of the total infections [37]. Other African countries that recorded high numbers of health worker infections included Algeria (11,936, 9.5%), Kenya (7542, 6.0%), Zimbabwe (5366, 4.3%), and Mozambique (4779, 3.8%). Only Eritrea has not reported any healthcare worker infections [37].

Identifying and understanding coexisting diseases associated with COVID-19 severity can help identify the potential risk factors and predictors of the disease severity, manage patients at risk, and develop policies and guidelines to effectively allocate healthcare resources that will reduce future risk of severe COVID-19 disease [6,38]. Estimating the prevalence of risk factors in this population group is important to guide occupational health measures to protect HCWs and maintain a functioning healthcare system. Mitigating the risk of COVID-19 infection among HCWs is a high priority; however, data to inform such initiatives are limited. Currently, studies from developing countries on COVID-19 and comorbidities among HCWs are limited to those from China. This has been mainly due to insufficient data collection during the pandemic in other developing countries [39]. Other studies in SA have reported risk factors for mortality, whilst this study examines risk factors for severe disease. A national COVID-19 hospital surveillance system has been locally established which captures HCW status. In this manuscript, we report on the prevalence of comorbidities among HCW COVID-19 admissions in South Africa and its association with disease severity (i.e., severe disease requiring intensive care or ventilation and possibly death). This study will provide information to guide recommendations for effective preventive measures to reduce transmission of the virus to other workers and patients, thus reducing secondary transmission to communities, as well.

## 2. Materials and Methods

### 2.1. Data Sources

The DATCOV national hospital surveillance system was established in March 2020 to monitor COVID-19 hospital admissions in South Africa. DATCOV contains data on HCWs across the nine provinces in SA that had a positive test for SARS-CoV-2, using real-time reverse-transcription polymerase chain reaction (rRT-PCR) assay or a positive SARS-CoV-2 antigen test, with a confirmed duration of stay in hospital of one full day or longer, regardless of age or reason for admission.

### 2.2. Study Population

This retrospective study design included HCWs who tested positive for SARS-CoV-2 who were admitted to public and private health facilities in South Africa from 5 March 2020 to 31 December 2021. HCW status included administrators and porters, doctors, nurses, allied healthcare workers, laboratory staff, and paramedics.

### 2.3. Data Analysis

Frequencies and percentages were used to summarize categorical variables. In the analysis, variables that reported on disease severity of the patients, such as HCWs admitted to intensive care unit (ICU) or high care unit (HCU), those who reported having acute respiratory distress syndrome (Ever ARDS), those who received supplemental oxygen and or were on ventilation, and those with mortality as an outcome status, were combined to create the dependent variable, disease severity. All patients who died of non-COVID causes were excluded from the analysis. Pearson chi-squared and Fishers exact tests were used to assess the association between study risk factors stratified by disease severity. A multivariable logistic regression model was implemented to assess the association of disease severity and existing comorbidities (hypertension, diabetes, chronic renal diseases, chronic pulmonary asthma, chronic cardiovascular diseases, current and past tuberculosis (TB), and HIV status), adjusting for age, sex, gender, and sector. A maximum likelihood test was used to include variables at the 5% level to generate a final model. The results were presented as unadjusted odds ratios (ORs) and adjusted odds ratios (aORs) and 95% confidence intervals. To investigate the effect of missing observation on the final model, a model with the variable containing more than 50% missing was fitted, which was compared to the model without the variable. Obesity was therefore excluded in the regression analysis, as it contained more than 50% missing data. The Hosmer–Lemeshow test for logistic regression was used for goodness of fit. A 5% significance level representing a *p* < 0.05 to reject the null hypothesis of no association was used. A data analysis was conducted by using STATA 15 (Stata Corp^®^, College Station, Texas, USA). We followed STROBE guideline recommendations.

## 3. Results

A cumulative total of 10,149 HCW admissions related to COVID-19 were reported across 666 hospitals, 407 from the public sector and 259 from the private sector, from 5 March 2020 to 31 December 2021 (Table 1). Table 1 shows the clinical and demographic characteristics of admitted HCWs reported with and without disease severity. This represented 3.8% (10,149/266,516) of all COVID-19 admissions. The majority of HCW admissions (83.6%) was recorded in four of the nine provinces, with the highest proportion reported in Gauteng (34.5%), followed by 23.6% in KwaZulu-Natal, 12.9% in Eastern Cape, and 12.6% in Western Cape provinces (Table 1). There were 2941 (29.0%) admissions in HCWs 50–59 years and 2088 (20.6%) in 60 years and older, comprising 50% of all admissions in this group (*p* < 0.001). A majority of COVID-19 admissions in HCWs were female (67.2%), and 2.5% were pregnant at the time. The most commonly reported comorbidity among HCWs admitted with disease severity was hypertension (42.7%)(*p* < 0.001), followed by diabetes (27.6%) (*p* < 0.001) and obesity (22.8%)(*p* < 0.001). Only 6.7% of HCWs reported having asthma (*p* = 0.811), HIV (5.1%) (*p* = 0.416), cardiac disease (2.5%), and TB (1.4%) (*p* = 0.117) as comorbidities.

The risk of disease severity among HCWs increased with age, with HCWs between 50 and 59 years having four times the risk of disease severity (aOR 4.3; 95% CI 3.4–5.4) and those older (≥60 years) with seven times the risk of disease severity (aOR 6.1; 95% CI 4.8–7.7) compared to HCWs in the younger age group (20–29 years) (Table 2). Male HCWs had a higher risk (aOR 1.5, 95% CI 1.4–1.7) of disease severity compared to female HCWs. Having the comorbidities hypertension (aOR 1.3; 95% CI 1.2–1.5), diabetes (aOR 1.3; 95% CI 1.2–1.5), and HIV (aOR 1.4; 95% CI 1.1–1.7) was significantly associated with increased risk of disease severity among HCWs. In the adjusted model, cardiac disease, asthma, TB (current/past), chronic kidney disease, and malignancy were not significantly associated with COVID-19 disease severity (Table 2). The risk of disease severity significantly increased among HCW admissions in Gauteng, Kwa-Zulu Natal, Mpumalanga, North West, Eastern Cape, Free State, and Limpopo provinces in South Africa compared to the Western Cape province (Table 2).

## 4. Discussion

This study is one of the largest studies in developing countries investigating 10,149 HCW admissions related to COVID-19 from 5 March 2020 to 31 December 2021. Although many countries in the WHO African region had seen a decline in COVID-19 cases for three consecutive months since May 2021, South Africa still ranked the highest [37]. Collectively, the country reported the highest number of COVID-19 cases in the region, with 3,559,230 cases (46.4%), followed by Ethiopia, with 457,322 (6.0%) cases [37]. The highest number of deaths in the region (93,364 deaths, 58.0% of all deaths) was also reported by South Africa [37]. With an increased number of HCW infections and an overwhelmed health system, determining the prevalence of risk factors among HCWs is critical to guide occupational health measures to protect frontline workers and stabilize the healthcare system’s human resources.

In this study, the risk of disease severity among HCWs increased with age, with HCWs in the older age group (≥60 years) having seven times the odds of disease severity compared to HCWs in the younger age group (20–29 years). The older age in most studies was associated with severe COVID-19 patients compared to COVID-19 in younger patients [29,30,32,40]. This poses a significant challenge, as it points to the loss of older HCWs who are more experienced and possess the much-needed skills to sustain the health system during a pandemic. Male HCWs were at an increased risk of severe disease compared to female HCWs. This is consistent with what has been reported from other studies [30,32,37]. Understanding the gender risk helps us to strategize resources, especially among male-dominated HCW teams. The risk of disease severity was significantly increased among HCW admissions in Gauteng, Kwa-Zulu Natal, Mpumalanga, and Limpopo provinces compared to the Western Cape. Incidentally, Gauteng and Kwa-Zulu Natal account for approximately 50% of the total number of confirmed COVID-19 cases, putting enormous strain on the healthcare system; this also accounts for a majority HCWs who are meant to care for the public requiring advanced medical care themselves.

These findings are important, as most previous COVID-19 studies have not reported the prevalence of comorbidities as a risk of developing severity of COVID-19 disease among HCWs. However, these studies primarily focused on COVID-19 and risk factors among the general public [6,41,42,43]. Only one study by Gibson and Greene (2020) investigated the risk of severe COVID-19 among HCWs who directly work with patients; the authors found that a quarter of the HCWs were deemed at higher risk of COVID-19 due to their multiple high-risk comorbidities. One of the noted comorbidities within the study was chronic lung disease/asthma attack in the past year [44]. However, the study by Gibson and Greene (2020) was not based on data acquired during the pandemic but on retrospective data regarding comorbidities among HCWs [44]. It does, however, highlight the importance of identifying vulnerable work populations. It is evident that HCW risk of occupational exposure to SARS-CoV-2 may vary from very high to high, medium, or low risk depending on the work activities they perform. In terms of the COVID-19 Direction on Health and Safety in the Workplace issued by the South African Ministry of Employment and Labour, employers must ensure that preventive measures are consistent with the overall national strategies and policies to minimize the spread of COVID-19 in the workplace [45]. The latter includes identifying vulnerable employees, that is, those with known or disclosed health issues or comorbidities or any other condition that may place the employee at a higher risk of complications or death than other employees if infected with COVID-19, or those above the age of 60 years, who are at a higher risk of complications or death if infected.

Furthermore, employers must take special measures to mitigate the risk of COVID-19 for vulnerable employees. However, there have been differing reports on the risk factors for COVID-19-related severity and mortality, with considerable variation in the prevalence estimates of comorbidities and their effects on COVID-19 severity. Risk stratification and effective control strategies for COVID-19 hinge on the risk factors, which may differ by geographical location and worker population. Numerous studies have identified comorbidities, including obesity, hypertension, diabetes, and HIV, with the severe infections and sometimes death of COVID-19 [29,31,32,40]. Asthma, for example, although a respiratory tract disease, was not associated with COVID-19 [46]. However, these focus on the general population and did not include HCWs from developing countries. In our study, adjusted analyses indicate that HCW comorbidities were significantly associated with disease severity and increased the risk of disease severity. These were hypertension (aOR 1.3; 95% CI 1.2–1.5), diabetes (aOR 1.3; 95% CI 1.2–1.5), and HIV (aOR 1.4; 95% CI 1.1–1.7). The HIV results were similar to those reported for patients with COVID-19 infection admitted to African high-care or intensive-care units (ACCCOS) with HIV/AIDS (1.91; 1.31–2.79). However, our risk ratio findings were marginally higher for hypertension and marginally higher for diabetes reported in the ACCCOS study. It is well-known that SARS-CoV-2 has overwhelmed healthcare systems globally by causing high rates of severe illness and more so if the rates of critical illness are high among the HCWs meant to provide care for others. Given that South Africa has a high burden of non-communicable disease, obesity, HIV, and TB [34], identifying worker vulnerabilities to mitigate the risk of developing severe COVID-19 and the loss of skilled HCWs is important. Furthermore, South Africa’s large untreated HIV population poses a risk to the generation of new viral variants [47], undermining prevention strategies to contain the pandemic to an endemic state. New variants evade pre-existing immunity and favor viral survival through enhanced transmissibility, resulting in acute surges with inherent hospitalization and subsequent severity of disease [48]. In addition, chronic diseases share several features with infectious disorders, including the pro-inflammatory state, as well as the attenuation of the innate immune response. For example, metabolic disorders may result in reduced immune function impairing macrophage and lymphocyte function, leading to diseases susceptibility [43]. Therefore, understanding the HCW exposure and risk for SARS-CoV-2 infection is important for characterizing the virus transmission patterns, risks factors for infection, and severity of disease and informing the effectiveness of infection prevention and control practices. Lastly, despite being the last alternative in the hierarchy of controls, wearing appropriate and well-fitted PPE is still one of the most effective preventative measures, particularly in poorly controlled areas and in settings where exposure is unknown or unpredictable, such as healthcare facilities [49].

## 5. Limitations

Data submitted to DATCOV are dependent on information submitted by healthcare institutions; thus, completeness of the data is a limitation in this study. The system has not yet been linked to other data sources, such as laboratory records, mortality records, and other national hospital records, to verify and confirm occupation and comorbid disease status; this may result in the under-reporting of such fields. Thus, we could not further classify HCWs according to occupation and job type. The DATCOV surveillance system also does not include vaccination status, and the effect of vaccination on severe disease was, therefore, not analyzed for this paper. Since HCW admissions are recorded retrospectively, it is possible that not all admissions from 5 March 2020 to 31 December 2021 were recorded. In addition, there may also be misclassification of HCW status, resulting in undercounting. The results of this study cannot be generalizable to HCW population in South Africa, as the analysis included admitted HCWs diagnosed with COVID-19 infection.

## 6. Conclusions

Age, gender, hypertension, diabetes, and HIV are strong predictors of the prognosis of COVID-19 in HCWs in SA. These findings can help in identifying the HCWs with the potential of developing severe COVID-19 disease. The information is important in the development of occupational health policies and vulnerability risk assessments for HCWs in light of future COVID-19 waves or similar outbreaks.

## Figures and Tables

**Table 1 ijerph-19-05519-t001:** Characteristics of admitted HCWs with severe and non-severe COVID-19 disease outcome, 5 March 2020 to 31 December 2021 (*N* = 10,149).

Characteristic	HCWs (*N* = 10,149)		Total (*N*, %)	Chi, *p*-Value
	Disease Severity (YES) (*n* = 5300; 52.2%)	Disease Severity (NO) (*n* = 4849; 47.8%)		
**Age Group (Years)**				<0.001
20–29	158 (3.0)	550 (11.3)	708 (7.0)	
30–39	692 (13.1)	1230 (25.4)	1922 (18.9)	
40–49	1202 (22.7)	1288 (26.6)	2490 (24.5)	
50–59	1769 (33.4)	1172 (24.2)	2941 (29.0)	
≥60	1479 (27.9)	609 (12.6)	2088 (20.6)	
**Sex**				<0.001
Female	3241 (61.2)	3583 (73.9)	6824 (67.2)	
Male	2059 (38.9)	1265 (26.1)	3324 (32.8)	
**Pregnancy (*n*/*N*, %)**				<0.001
No	5264 (99.3)	4630 (95.5)	9894 (97.5)	
Yes	36 (0.7)	219 (4.5)	255 (2.5)	
**Chronic Diseases (*n*/*N*, %)**				
**Hypertension**				<0.001
No	2740/4781 (57.3)	2634/3651 (72.1)	5374/8432 (63.7)	
Yes	2041/4781 (42.7)	1017/3651 (27.9)	3058/8432 (36.3)	
**Diabetes**				<0.001
No	3404/4703 (72.4)	2962/3600 (82.3)	6366/8303 (76.7)	
Yes	1299/4703 (27.6)	638/3600 (17.7)	1937/8303 (23.3)	
**Cardiac Disease**				<0.001
No	4482/4596 (97.5)	3516/3559 (98.8)	7998/8155 (98.1)	
Yes	114/4596 (2.5)	43/3559 (1.2)	157/8155 (1.9)	
**Asthma**				0.811
No	3365/3607 (93.3)	4306/4630 (93.0)	7671/8237 (93.1)	
Yes	242/3607 (6.7)	324/4630 (7.0)	566/8237 (6.9)	
**Obesity**				<0.001
No	1005/1302 (77.2)	1076/1195 (90.0)	2081/2497 (83.3)	
Yes	297/1302 (22.8)	119/1195 (10.0)	416/2497 (16.7)	
**HIV**				0.416
No	4336/4570 (94.9)	3366/3564 (94.4)	7702/8134 (94.7)	
Yes	234/4570 (5.1)	198/3564 (5.6)	432/8134 (5.3)	
**TB (Current/Past)**				0.117
No	4530/4596 (98.6)	3520/3582 (98.3)	8050/8178 (98.4)	
Yes	66/4596 (1.4)	62/3582 (1.7)	128/8178 (1.6)	
**Chronic Kidney Disease**				0.837
No	4552/4582 (99.4)	3532/3554 (99.4)	8084/8136 (99.4)	
Yes	30/4528 (0.7)	22/3554 (0.6)	52/8136 (0.6)	
**Malignancy**				0.929
No	4554/4574 (99.6)	3515/3533 (99.5)	8069/8107 (99.5)	
Yes	20/4574 (0.4)	18/3533 (0.5)	38/8107 (0.5)	
**Province**				<0.001
Gauteng	1739 (32.8)	1763 (36.4)	3502 (34.5)	
Kwa-Zulu Natal	1250 (23.6)	1149 (23.7)	2399 (23.6)	
Eastern Cape	793 (15.0)	512 (10.6)	1305 (12.9)	
Western Cape	547 (10.3)	736 (15.2)	1283 (12.6)	
North West	407 (7.7)	286 (5.9)	639 (6.8)	
Free State	159 (3.0)	192 (4.0)	351 (3.5)	
Mpumalanga	186 (3.5)	94 (1.9)	280 (2.8)	
Limpopo	163 (3.1)	69 (1.9)	232 (2.3)	
Northern Cape	56 (1.1)	48 (1.0)	104 (1.0)	

**Table 2 ijerph-19-05519-t002:** Factors associated with disease severity among HCWs admitted with COVID-19 (*N* = 10 149), 5 March 2020–31 December 2021.

Characteristic	Disease Severity HCWs	Unadjusted OR (95% CI)	*p*	Adjusted OR (95% CI)	*p*
**Age Group (Years)**					
20–29	158 (3.0)	ref (1.00)		ref	
30–39	692 (13.1)	2.0 (1.6–2.4)	<0.001	1.9 (1.5–2.3)	<0.001
40–49	1202 (22.7)	3.2 (2.7–3.9)	<0.001	3.1 (2.5–4.0)	<0.001
50–59	1769 (33.4)	5.3 (4.3–6.4)	<0.001	4.3 (3.4–5.4)	<0.001
≥60	1479 (27.9)	8.4 (6.9–10.3)	<0.001	6.1 (4.8–7.7)	<0.001
**Sex**					
Female	3241 (61.2)	ref		ref	
Male	2059 (38.9)	1.8 (1.7–2.0)	<0.001	1.5 (1.4–1.7)	0.003
**Chronic Diseases**					
**Hypertension**					
No	2740/4781 (57.3)	ref		ref	
Yes	2041/4781 (42.7)	1.9 (1.8–2.1)	<0.001	1.3 (1.2–1.5)	0.027
**Diabetes**					
No	3404/4703 (72.4)	ref		ref	
Yes	1299/4703 (27.6)	1.8 (1.6–2.0)	<0.001	1.3 (1.2–1.5)	<0.001
**Cardiac Disease**					
No	4482/4596 (97.5)	ref			
Yes	114/4596 (2.5)	2.1 (1.5–3.0)	<0.001	1.4 (0.9–2.0)	0.133
**Asthma**					
No	3365/3607 (93.3)	ref		ref	
Yes	242/3607 (6.7)	1.0 (0.9–1.2)	0.607		
**HIV**					
No	4336/4570 (94.9)	ref		ref	
Yes	234/4570 (5.1)	0.9 (0.8–1.1)	0.385	1.4(1.1–1.7)	0.008
**TB (Current/Past)**					
No	4530/4596 (98.6)	ref			
Yes	66/4596 (1.4)	0.8 (0.6–1.2)	0.287		
**Chronic Kidney Disease**					
No	4552/4582 (99.4)	ref			
Yes	30/4528 (0.7)	1.1 (0.6–1.8)	0.841		
**Malignancy**					
No	4554/4574 (99.6)	ref			
Yes	20/4574 (0.4)	0.9 (0.4–1.6)	0.929		
**Province**					
Western Cape	1739 (32.8)	ref		ref	
Gauteng	1250 (23.6)	1.3 (1.2–1.5)	<0.001	2.2 (1.8–2.5)	<0.001
Kwa-Zulu natal	793 (15.0)	1.5 (1.3–1.7)	<0.001	1.6 (1.4–1.9)	<0.001
Eastern Cape	547 (10.3)	2.1 (1.8–2.4)	<0.001	2.2 (1.8–2.6)	<0.001
North West	407 (7.7)	1.9 (1.6–2.3)	<0.001	2.3 (1.8–2.8)	<0.001
Free State	159 (3.0)	1.1 (0.9–1.4)	0.372	1.4 (1.0–1.8)	0.030
Mpumalanga	186 (3.5)	2.7 (2.0–3.5)	<0.001	2.3 (1.8–2.8)	<0.001
Limpopo	163 (3.1)	3.2 (2.3–4.3)	<0.001	2.8 (1.8–4.4)	<0.001
Northern Cape	56 (1.1)	1.6 (1.1–2.3)	0.028	1.3 (0.8–2.2)	0.304
**Hosmer–Lemeshow chi^2^ (8) Prob > chi^2^ = 0.0665**					

## Data Availability

Data are available upon reasonable request and within the prescripts of the Protection of Personal Information Act (POPIAct).

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
