# Peer review of "Disease Severity and Comorbidities among Healthcare Worker COVID-19 Admissions in South Africa: A Retrospective Analysis"

_ijerph, 2022, doi:10.3390/ijerph19095519_

Round 1

Reviewer 1 Report

The authors presented a study on the prevalence of comorbidities among the South African healthcare worker with COVID-19 and its association with disease severity. This manuscript is well written and the methodology and results are clearly presented. With such a large sample size, I believe the finding of this study will provide important insights into the subject of the study. I have only a minor comment on the abstract where the first sentence needs to be revised since this is a strong statement without supporting data.

Author Response

REVIEWER 1

I have only a minor comment on the abstract where the first sentence needs to be revised since this is a strong statement without supporting data.

RESPONSE:

The first sentence has been amended to “Healthcare workers (HCWs) are among the most vulnerable in contracting severe acute respiratory syndrome coronavirus 2 (SARS-CoV-2) infection.

Reviewer 2 Report

The paper entitled Disease severity and comorbidities among Healthcare worker 2 COVID-19 admissions in South Africa. by  Edith Ratshikhopha et al. is an informative article about COVID 19, extracting results by using  big sample and it could be published. For more extensive study for the correlation among different disease with Covid 19, the authors could use other techniques such as complex networks. Also, the author can give the main results from tables 1 and 2 using graphs.

Author Response

Dear Reviewer,

RESPONSE:

The authors acknowledge the comment and agree with the reviewer that using Directed Acyclic Graphs can assist with making inference on study variables.  However, we have described comorbidities assessed in the study as well known risk factors for COVID-19 disease severity.  Where,

Table 1 presents a descriptive analyses or summary of the study population stratified by outcome (disease severity in this case) and,

Table 2 presents the bivariate and multivariable analysis of risk factors with the outcome which are presented as unadjusted and adjusted ODDS ratio, which cannot be presented in a graphical format.

Therefore, table 1 and table 2 cannot be presented using graphs.

Reviewer 3 Report

This study identified risk factors for disease severity among health workers infected with covid. The manuscript is well-written. 

Between March 2020 to December 2021, covid vaccinations get available. If there is data on covid vaccinations, please consider to include the vaccination status.  In addition (even if individual vaccination status is not available), a stratified analysis by before and after the time point of vaccine administration to HCW in South Africa will be more practically informative.

Author Response

Dear Reviewer,

RESPONSE:

Thank you for the comment which would add value to the paper, unfortunately vaccination status was not available for this analysis. 

The main objective of the paper is to assess association between disease severity and comorbidities among HCWs admitted with COVID-19. In a future publication, if the data becomes available, the authors may explore the impact of vaccination on COVID-19 outcomes. Added to limitations:  The DATCOV surveillance system also does not include vaccination status and the effect of vaccination on severe disease was therefore not analysed for this paper.

Reviewer 4 Report

  • Modify the title in Disease severity and comorbidities among Healthcare worker COVID-19 admissions in South Africa: a retrospective analysis.

Introduction

  • line , The worldwide outbreak of coronavirus disease-19 (COVID-19) has already put healthcare workers (HCWs) at a high risk of infection. The question of how to give HCWs the best protection against infection is a priority.  Covering more of the body, and a higher-level specification of masks and respirators provided better protection for HCWs. A community mask could be beneficial.  please discuss  this precedent review and cite doi:10.1111/ijcp.13617.
  • line 37, The pandemic caused by SARS-CoV2 has stressed health care systems worldwide, with a higher volume of patients, combined with an increased need for intensive care and potential transmission, which has forced the reorganization of hospitals and care delivery models. Several approaches to minimize risk to Otolaryngologists during their patients infected with COVID-19 care have been performed. Standard operating procedures have been adapted both for facilities and for health care workers, including the development of well-defined and segregated patient care areas for treating those affected by COVID-19. Personal protective equipment (PPEs) availability and adequate healthcare provider training on their use should be ensured. Preventive measures are especially important in Otolaryngology-Head and Neck Surgery, as the exposure to saliva suspensions, droplets, and aerosols are increased in the upper aero-digestive tract routine examination. Moreover, the frequent invasive procedures, such as laryngoscopy, intubation, or tracheotomy placement and care, represent a high risk of contracting COVID-19. please discuss.
  • line 40, An interesting paper discussed the only efficient means of protecting and mitigating infectious contagion has been the use of personal protective equipment, especially by healthcare workers. However, masks affect the humidification process of inhaled air, possibly leading to a basal inflammatory state of the upper airways. A higher prevalence of nasal and ocular symptoms, perceived reduced work performance, difficulty in concentrating, and sleep disorders were found. After two weeks of adhering to a list of good practices that we recommended, significant reversibility of the symptoms was investigated, and work performance enhancement was observed. 

Methods

  • add the strobe and equator guidelines
  • which study design?
  • committee code?
  • statistical analysis p value?

Results

  • table I add chi square or t test in table

Discussion

  • discuss with in the adjusted model, cardiac disease, asthma, TB (current/past), chronic kidney disease, and malignancy were not significantly associated with COVID-19 disease severity. as possible study limitations
  • line 176, Nucleic acid testing is the standard method for the diagnosis of viral infections. To increase the sensitivity of COVID-19 diagnoses recently were developed an IgM-IgG combined assay and tested in patients with suspected SARS-CoV-2 infection. In total, 56 patients were enrolled in this study, and SARS-CoV-2 was detected by using both IgM-IgG antibody and nucleic acid tests. Clinical and laboratory data were collected and analyzed. Patients who develop severe illness might experience longer virus exposure times and develop a more severe inflammatory response. The IgM-IgG test is an accurate and sensitive diagnostic method. A combination of nucleic acid and IgM-IgG testing is a more sensitive and accurate approach for the diagnosis and early treatment of COVID-19. please discuss and cite doi:10.1002/jmv.25930
  • Understanding the humoral responses to severe acute respiratory syndrome coronavirus 2 (SARS-CoV-2) is critical to improving diagnostics, therapy and vaccines. Deep serological profiling of 232 coronavirus disease 2019 (COVID-19) patients and 190 pre-COVID-19 era controls using VirScan revealed more than 800 epitopes in the SARS-CoV-2 proteome, including 10 epitopes likely recognized by the neutralizing antibodies. Pre-existing antibodies in controls recognized SARS-CoV-2 ORF1, while only the patient's COVID-19 antibodies recognized primarily the spike protein and nucleoprotein. A machine learning model trained on VirScan data predicted SARS-CoV-2 exposure history with 99% sensitivity and 98% specificity; Rapid Luminex-based diagnostics have been developed from the most discriminatory SARS-CoV-2 peptides. Individuals with more severe COVID-19 showed stronger and broader SARS-CoV-2 responses, weaker antibody responses to previous infections, and a higher incidence of cytomegalovirus and herpes simplex 1 virus, possibly influenced by demographic covariates. Among hospitalized patients, males produce stronger SARS-CoV-2 antibody responses than females. please cite doi:10.1126/science.abd4250

Add a study limitation section

Author Response

Dear Reviewer,

Reviewer Comments

  1. Modify the title in Disease severity and comorbidities among Healthcare worker COVID-19 admissions in South Africa: a retrospective analysis.

RESPONSE:

The title has been amended as per suggestion.

  1. The worldwide outbreak of coronavirus disease-19 (COVID-19) has already put healthcare workers (HCWs) at a high risk of infection. The question of how to give HCWs the best protection against infection is a priority. Covering more of the body, and a higher-level specification of masks and respirators provided better protection for HCWs. A community mask could be beneficial.  please discuss this precedent review and cite doi:10.1111/ijcp.13617.

RESPONSE:

This was addressed by adding the following in lines 254-257: “Lastly, despite being the last alternative in the hierarchy of controls, wearing appropriate and well fitted PPE is still one of the most effective preventative measures, particularly in poorly controlled areas and in settings where exposure is unknown or unpredictable like healthcare facilities.”

  1. line 37, The pandemic caused by SARS-CoV2 has stressed health care systems worldwide, with a higher volume of patients, combined with an increased need for intensive care and potential transmission, which has forced the reorganization of hospitals and care delivery models. Several approaches to minimize risk to Otolaryngologists during their patients infected with COVID-19 care have been performed. Standard operating procedures have been adapted both for facilities and for health care workers, including the development of well-defined and segregated patient care areas for treating those affected by COVID-19. Personal protective equipment (PPEs) availability and adequate healthcare provider training on their use should be ensured. Preventive measures are especially important in Otolaryngology-Head and Neck Surgery, as the exposure to saliva suspensions, droplets, and aerosols are increased in the upper aero-digestive tract routine examination. Moreover, the frequent invasive procedures, such as laryngoscopy, intubation, or tracheotomy placement and care, represent a high risk of contracting COVID-19. please discuss.

RESPONSE:

Thank you for the comment and we do agree that the pandemic has stressed healthcare systems worldwide forcing a reorganization of hospitals and care delivery models. 

  1. line 40, An interesting paper discussed the only efficient means of protecting and mitigating infectious contagion has been the use of personal protective equipment, especially by healthcare workers. However, masks affect the humidification process of inhaled air, possibly leading to a basal inflammatory state of the upper airways. A higher prevalence of nasal and ocular symptoms, perceived reduced work performance, difficulty in concentrating, and sleep disorders were found. After two weeks of adhering to a list of good practices that we recommended, significant reversibility of the symptoms was investigated, and work performance enhancement was observed. 

RESPONSE:

Thank you for the comment.

  1. add the strobe and equator guidelines

RESPONSE:

We did follow strobe guidelines and have added this for each section where the item was addressed. 

Item

Recommendation

Reported on manuscript page

Title and abstract

1

(a) Indicate the study’s design with a commonly used term in the title or the abstract

(b) Provide in the abstract an informative and balanced summary of what was done and what was found

1,3

1

Introduction

Background/rationale

2

Explain the scientific background and rationale for the investigation being reported

2-3

Objectives

3

State specific objectives, including any prespecified hypotheses

3

Methods

Study design

4

Present key elements of study design early in the paper

3

Setting

5

Describe the setting, locations, and relevant dates, including periods of recruitment, exposure, follow-up, and data collection

3

Participants

6

(a) Cross-sectional study—give the eligibility criteria, and the sources and methods of selection of participants

n/a

Variables

7

Clearly define all outcomes, exposures, predictors, potential confounders, and effect modifiers. Give diagnostic criteria, if applicable

3

Data sources/

measurement

8*

For each variable of interest give sources of data and details of methods of assessment (measurement). Describe comparability of assessment methods if there is more than one group

3

Bias

9

Describe any efforts to address potential sources of bias

8

Study size

10

Explain how the study size was arrived at

4

Quantitative variables

11

Explain how quantitative variables were handled in the analyses. If applicable, describe which groupings were chosen, and why

3

Statistical methods

12

(a) Describe all statistical methods, including those used to control for confounding

(b) Describe any methods used to examine subgroups and interactions

(c) Explain how missing data were addressed

(d) Cross-sectional study—if applicable, describe analytical methods taking account of sampling strategy

(e) Describe any sensitivity analyses

3

Results

Participants

13*

(a) Report the numbers of individuals at each stage of the study—eg, numbers potentially eligible, examined for eligibility, confirmed eligible, included in the study, completing follow-up, and analysed

(b) Give reasons for non-participation at each stage

(c) Consider use of a flow diagram

4

Descriptive data

14*

(a) Give characteristics of study participants (eg, demographic, clinical, social) and information on exposures and potential confounders

(b) Indicate the number of participants with missing data for each variable of interest

4

Outcome data

15*

Cross-sectional study—report numbers of outcome events or summary measures

n/a

Main results

16

(a) Give unadjusted estimates and, if applicable, confounder-adjusted estimates and their precision (eg, 95% confidence interval). Make clear which confounders were adjusted for and why they were included

(b) Report category boundaries when continuous variables were categorised

(c) If relevant, consider translating estimates of relative risk into absolute risk for a meaningful time period

5-6

Other analyses

17

Report other analyses done—eg, analyses of subgroups and interactions, and sensitivity analyses

56

Discussion

Key results

18

Summarise key results with reference to study objectives

6-8

Limitations

19

Discuss limitations of the study, taking into account sources of potential bias or imprecision. Discuss both direction and magnitude of any potential bias

8

Interpretation

20

Give a cautious overall interpretation of results considering objectives, limitations, multiplicity of analyses, results from similar studies, and other relevant evidence

8-9

Generalisability

21

Discuss the generalisability (external validity) of the study results

8

Other information

Funding

22

Give the source of funding and the role of the funders for the present study and, if applicable, for the original study on which the present article is based

9

  1. which study design?

RESPONSE:

The retrospective study design was included in the methods section.

  1. committee code?

RESPONSE:

We did not understand the reviewer’s suggestion but would be happy to address if further clarity was provided.  If it relates to ethics committee code then “Ethical approval for this study was obtained from the University of the Witwatersrand Human Research Ethics Committee (M160667).”.

  1. statistical analysis p value?

RESPONSE:

Addressed.  A 5% significance level representing a p < 0.05 to reject the null hypothesis of no association was used.

  1. table I add chi square or t test in table

RESPONSE:

Addressed.  We used a chi-square test for categorical variables.

  1. discuss with in the adjusted model, cardiac disease, asthma, TB (current/past), chronic kidney disease, and malignancy were not significantly associated with COVID-19 disease severity. as possible study limitations

RESPONSE:

Thank you for the comment.  We have stated in the limitations that the study findings could be affected by incomplete or poor quality data.

  1. line 176, Nucleic acid testing is the standard method for the diagnosis of viral infections. To increase the sensitivity of COVID-19 diagnoses recently were developed an IgM-IgG combined assay and tested in patients with suspected SARS-CoV-2 infection. In total, 56 patients were enrolled in this study, and SARS-CoV-2 was detected by using both IgM-IgG antibody and nucleic acid tests. Clinical and laboratory data were collected and analyzed. Patients who develop severe illness might experience longer virus exposure times and develop a more severe inflammatory response. The IgM-IgG test is an accurate and sensitive diagnostic method. A combination of nucleic acid and IgM-IgG testing is a more sensitive and accurate approach for the diagnosis and early treatment of COVID-19. please discuss and cite doi:10.1002/jmv.25930

RESPONSE:

Thank you for the comment and sharing the publication.

  1. Understanding the humoral responses to severe acute respiratory syndrome coronavirus 2 (SARS-CoV-2) is critical to improving diagnostics, therapy and vaccines. Deep serological profiling of 232 coronavirus disease 2019 (COVID-19) patients and 190 pre-COVID-19 era controls using VirScan revealed more than 800 epitopes in the SARS-CoV-2 proteome, including 10 epitopes likely recognized by the neutralizing antibodies. Pre-existing antibodies in controls recognized SARS-CoV-2 ORF1, while only the patient's COVID-19 antibodies recognized primarily the spike protein and nucleoprotein. A machine learning model trained on VirScan data predicted SARS-CoV-2 exposure history with 99% sensitivity and 98% specificity; Rapid Luminex-based diagnostics have been developed from the most discriminatory SARS-CoV-2 peptides. Individuals with more severe COVID-19 showed stronger and broader SARS-CoV-2 responses, weaker antibody responses to previous infections, and a higher incidence of cytomegalovirus and herpes simplex 1 virus, possibly influenced by demographic covariates. Among hospitalized patients, males produce stronger SARS-CoV-2 antibody responses than females. please cite doi:10.1126/science.abd4250

RESPONSE:

Thank you for the comment and sharing the publication.

  1. Add a study limitation section

RESPONSE:

This is included.